# Learning from Group Comparisons: Exploiting Higher Order Interactions

**Yao Li**
Department of Statistics
University of California, Davis
yaoli@ucdavis.edu

**Minhao Cheng**
Department of Computer Science
University of California, Los Angeles
mhcheng@ucla.edu

**Kevin Fujii**
Department of Statistics
University of California, Davis
kmfujii@ucdavis.edu

**Fushing Hsieh**
Department of Statistics
University of California, Davis
fhsieh@ucdavis.edu

**Cho-Jui Hsieh**
Department of Computer Science
University of California, Los Angeles
chohsieh@cs.ucla.edu

## Abstract

We study the problem of learning from group comparisons, with applications in predicting outcomes of sports and online games. Most of the previous works in this area focus on learning individual effects—they assume each player has an underlying score, and the "ability" of the team is modeled by the sum of team members' scores. Therefore, current approaches cannot model deeper interaction between team members: some players perform much better if they play together, while some players perform poorly together. In this paper, we propose a new model that takes the player-interaction effects into consideration. However, under certain circumstances, the total number of individuals can be very large, and number of player interactions grows quadratically, which makes learning intractable. In this case, we propose a latent factor model, and show that the sample complexity of our model is bounded under mild assumptions. Finally, we show that our proposed models have much better prediction power on several E-sports datasets, and furthermore can be used to reveal interesting patterns that cannot be discovered by previous methods.

## 1 Introduction

Nowadays there are a lot of online games in the form of group comparisons, and this e-sports industry is growing at an unexpected pace. For example, League of Legends (LoL) has attracted more than 11 million active players in each month; Dota 2 had a grand prize of near 25 million dollars last year. A big crowd of players and matches certainly creates many challenges: for instance, how to design a good matchmaking system to match two teams with similar strengths, and how to form a better team composition to win the game. To answer these questions, we consider the core problem of modeling group comparisons: given the results of previous games (each game is a group comparison between two teams), how to predict the outcome of an unseen game?

All the previous work in this area focuses on the individual scoring model, that is, assuming each player has an underlying score, and the "ability" of the team is modeled by the sum of team members'

scores. Through the process, one can also rank a player by his or her score in the model. For example, [13] extends the Bradley-Terry model to the group comparison setting; [12] proposed a TrueSkill algorithm to learn individual scores using a Bayesian model, which has now been used by game companies and sport analysts.

However, a common weakness of previous methods is that they ignore the player-interaction effects. In a team challenge, the players work together and will always influence each other, and this player-interaction effect can significantly alter game results. To make the prediction more accurate, incorporating the player-interaction effects are demanding. On the other hand, people are also interested in the cooperation effects between players. Team coach can pair a better team based on both individual abilities and cooperation abilities; game designers such as Blizzard can use the results to design their heroes. This brings us to the questions we wish to answer in this paper:

- Can incorporating player-interaction effects improve the prediction accuracy of the model? How to interpret those effects?

- If the total number of players is too large, how can our algorithm scale up and meanwhile maintain good generalization error and efficient computational time?

To answer the first question, we propose a new model that can incorporate pairwise effects, and show that the pairwise effects can be learned when there are not too many players. The player-interaction can not be fully modeled by pairwise effects. This is the first step, and investigating effects with order higher than two is our future work. As for the second question, we propose a latent factor model to describe pairwise interactions between players, and propose an efficient stochastic gradient descent algorithm to solve it. A theoretical bound of the sample complexity is provided under mild conditions.

In the experimental part, we test our model on online game datasets and show that our proposed models have much higher prediction accuracy than previous individual-score based models. For example, in Heroes of the Storm data our new models can get around 80% accuracy, while state-of-the-art models such as Trueskill can only achieve 60% accuracy.

## 2  Problem Setting

Assume there are $n$ individuals $\{1, \cdots, n\}$, and $T$ observed comparisons. Each game involves two disjoint teams $I_t^+$ and $I_t^-$, each of them is a subset of $\{1, \cdots, n\}$, indicating the players involved in the team. Without loss of generality we assume team $I_t^+$ wins the game, and team $I_t^-$ loses the game. For simplicity, we assume each game has an equal number of players on each team, and there could be $N = \binom{n}{L}$ different combinations, where $L = |I_t^+| = |I_t^-|$ is the number of players on each team. For each game, the outcome $o_t$ can be observed under two scenarios:

- Measured outcomes (scores): for each comparison, the *value* of the score difference is observed: $o_t \in \mathbb{R}$ can be any real number.
- Binary indicator outcomes (wins/losses): for each comparison, the *sign* of the score difference is revealed: $o_t \in \{+1, -1\}$.

Most problems are given in the form of second case. However, in some cases it is possible to observe the scores. For example, the score in an NBA game, or the number of kills in an online matching game. Our proposed approaches work for both cases since we assume a general loss function, while in the experiment we focus on binary outcomes.

## 3  Related Work

**Learning individual scores from group comparisons.**  Most of the previous work focus on learning individual scores by group comparisons [13, 12]. All of them make the following assumption:

**Assumption 1.** *The team's score is the sum over team members' scores:* $s_t^+ = \sum_{j \in I_t^+} w_j$, *where* $w_j$ *is the ability of player $j$. The observed outcome is determined by $s_t^+ - s_t^-$.*

For example, [13] proposed a generalized Bradley-Terry model: assume $w_j$ is the score for the $j$-th player, and

$$P(I_t^+ \text{ beats } I_t^-) = \exp(\sum_{j \in I_t^+} w_j) / \left( \exp(\sum_{j \in I_t^+} w_j) + \exp(\sum_{j \in I_t^-} w_j) \right),$$

then the MLE estimator for the underlying scores $\boldsymbol{w} \in \mathbb{R}^n$ can be estimated by $\hat{\boldsymbol{w}} = \arg\min_{\boldsymbol{w}} \sum_{t=1}^T \log P(I_t^+ \text{ beats } I_t^-)$. Trueskill [12] is a Bayesian approach for learning the scores using a similar generating model, and is used in most of the real world online game matching systems.

Here we also consider a simple but effective individual-score based method:

$$\min_{\boldsymbol{w} \in \mathbb{R}^n} \sum_{t=1}^T \ell(\boldsymbol{w}^T \boldsymbol{x}_t, o_t) + R(\boldsymbol{w}). \tag{1}$$

$\boldsymbol{w} \in \mathbb{R}^n$ is the individual score vector we want to learn. $\boldsymbol{x}_t \in \mathbb{R}^n$ is the indicator vector, where $(\boldsymbol{x}_t)_j = 1$ if $j \in I_t^+$, $(\boldsymbol{x}_t)_j = -1$ if $j \in I_t^-$, and $(\boldsymbol{x}_t)_j = 0$ for all other elements. Although we cannot find this simple model in the literature, in practice we found this often outperforms Trueskill and Bradley-Terry model when $\ell(\cdot, \cdot)$ is logistic loss and $R(\boldsymbol{w})$ is L2 regularization, so we also include this model in our comparisons in the experimental part. For all the individual score models, it is not hard to observe that they require at least $O(n)$ games in order to recover $n$ individual scores with small error.

**Factorization machine.** Factorization machines were introduced by Rendle [17]. They hold great promise in the applications with sparse predictors, especially when pairwise interaction of variables is useful and linear complexity with polynomial results is desired. For example, [16] introduced a factorized sparse model to identify high-order feature interactions in linear and logistic regression models. In this paper, we propose a factorization model to help scale up when number of players ($n$) is large.

**Other related work.** Ranking individuals from pairwise comparisons have been extensively studied. The famous Elo system [11] has been used for a long time for chess and other sports ratings. [19, 10] also proposed some different approaches with theoretical guarantee. [3, 4] recently provide a novel view of the ranking problem by modeling intransitivity in pairwise comparisons (intransitivity means $a > b, b > c, c > a$). However, all these papers consider pairwise comparisons, while we consider the problem of group comparisons in this paper.

Another recent line of research studies how to improve the ranking algorithm by exploiting feature information. [20] discussed a Bradley-Terry model with features. [21] applied a factorization model to incorporate feature information. [5] proposed a simple method to combine feature-based and comparison-based approaches and demonstrated the use of feature can reduce the complexity in theory. We do not consider features in this paper. However, since most of the online game matching data has features associated with each game, it is our future work to explore this area.

## 4 Exploiting higher order information

All the current approaches cannot model the pairwise relationships of team members: some players perform much better if they play together, and some players perform poorly together. We propose the following methods to model pairwise interactions.

**Basic Model for Higher Order Interactions.** We assume each player has its individual score $w_{jj}$. And for each pair of players, there is a pairwise score $w_{jq}$. A team's ability is modeled by

$$s_t^+ = \sum_{j,q \in I_t^+} w_{jq}.$$

Our goal is to learn the model so that the score $s_t^+$ is larger than $s_t^-$ for each game. Assume $\boldsymbol{e}_t^+, \boldsymbol{e}_t^- \in \{0,1\}^n$ are the indicator vectors for $I_t^+$ and $I_t^-$ respectively, where $(\boldsymbol{e}_t^+)_j = 1$ if $j \in I_t^+$ and $(\boldsymbol{e}_t^-)_j = 1$ if $j \in I_t^-$. Then the objective function can be written as

$$\text{Basic HOI:} \quad \min_{W \in \mathbb{R}^{n \times n}} \sum_{t=1}^T \ell((\boldsymbol{e}_t^+)^T W (\boldsymbol{e}_t^+) - (\boldsymbol{e}_t^-)^T W (\boldsymbol{e}_t^-), o_t) + \lambda \|W\|_F^2. \tag{2}$$

$W = (w_{jq}) \in \mathbb{R}^{n \times n}$ is the score matrix of players, where diagonal element $w_{jj}$ corresponds to the ability score of player $j$ and $w_{jq}$ corresponds to the pairwise score of players $(j, q)$. One way to solve (2) is by transforming it to classical empirical risk minimization. Since

$$(\boldsymbol{e}_t^+)^T W (\boldsymbol{e}_t^+) = \mathrm{tr}(W \boldsymbol{e}_t^+ (\boldsymbol{e}_t^+)^T) = \mathrm{vec}(W)^T \mathrm{vec}(\boldsymbol{e}_t^+ (\boldsymbol{e}_t^+)^T),$$

problem (2) is equivalent to

$$\min_{\boldsymbol{w} \in \mathbb{R}^{n^2}} \sum_{t=1}^{T} \ell(\boldsymbol{w}^T \boldsymbol{x}_t, o_t) + \lambda \|\boldsymbol{w}\|_2^2, \tag{3}$$

where $\boldsymbol{w} = \mathrm{vec}(W)^T, \boldsymbol{x}_t = \mathrm{vec}(\boldsymbol{e}_t^+ (\boldsymbol{e}_t^+)^T - \boldsymbol{e}_t^- (\boldsymbol{e}_t^-)^T)$. After this reformulation, (3) can be solved by standard SVM or logistic regression packages when $\ell(\cdot, \cdot)$ is hinge loss or logistic loss.

Indeed, this model is quite flexible and can be extended to extract higher order interactions, such as interactions among any 3 players, or 4 players. The only problem is the number of parameters will be very large when higher order information is used.

**Difficulty in scaling to large problems.** Our basic model is quite effective when the number of players $n$ is small (see our experimental results). However, in many real world problems $n$ is very large. For example, even a small online game would have tens of thousands of players, and popular games such as LoL or Heroes of the Storm typically have millions of players. Unfortunately, our basic model cannot scale to large $n$ due to the following two reasons:

- In terms of sample complexity, (2) has $n^2$ parameters. Based on standard statistical learning theory, it requires at least $O(n^2)$ observed samples to recover the underlying scores. Even for 10,000 players, (2) will require 100 million games to get a good estimate.
- In terms of computing, (2) requires $O(n^2)$ memory to store the $W$ matrix, which is typically dense unless making further structural assumption. Therefore, a standard solver will be hard to scale beyond $30,000$ players.

### 4.1 Factorization Model for Higher Order Interactions (Factorization HOI)

To overcome the large $n$ problem, we propose the following Factorization HOI model, which assumes a team's score can be written as

$$s_t^+ = \sum_{j \in I_t^+} w_j + \sum_{j \in I_t^+} \sum_{q \in I_t^+} \boldsymbol{v}_j^T \boldsymbol{v}_q.$$

Model parameters that have to be estimated are $\boldsymbol{w} \in \mathbb{R}^n$ and $V \in \mathbb{R}^{k \times n}$, each $\boldsymbol{v}_j$ is the $j$-th column of $V$. The hyper-parameter $k$ defines the dimensionality of the factorization.

In this model, we capture the individual strength by $w_j$, and each pairwise strength is modeled by $w_{jq} \approx \boldsymbol{v}_j^T \boldsymbol{v}_q$. This assumption is the key point which allows high quality and efficient parameter estimation of higher order interactions. An intuitive explanation about this model is that each player is associated with $k$ latent features, and the interaction between two players is modeled by the interaction of them via these latent features.

To estimate the parameters for Factorization HOI, we solve the following optimization problem:

$$\underset{\boldsymbol{w} \in \mathbb{R}^n, V \in \mathbb{R}^{k \times n}}{\operatorname{argmin}} \sum_{t=1}^{T} \ell(s_t^+ - s_t^-, o_t) + \frac{\lambda_w}{2} \|\boldsymbol{w}\|_2^2 + \lambda_V \|V\|_F^2 \tag{4}$$

$$= \underset{\boldsymbol{w} \in \mathbb{R}^n, V \in \mathbb{R}^{k \times n}}{\operatorname{argmin}} \sum_{t=1}^{T} \ell(\boldsymbol{w}^T (\boldsymbol{e}_t^+ - \boldsymbol{e}_t^-) + \sum_{j,q \in I_t^+} \boldsymbol{v}_j^T \boldsymbol{v}_q - \sum_{j,q \in I_t^-} \boldsymbol{v}_j^T \boldsymbol{v}_q, o_t)$$

$$+ \frac{\lambda_w}{2} \|\boldsymbol{w}\|_2^2 + \lambda_V \|V\|_F^2,$$

where $\lambda_w$ and $\lambda_V$ are the regularization parameters.

**Efficient Solver.** To solve (4), we propose the following algorithm that alternatively updates $\boldsymbol{w}$ and $V$. When $V$ is fixed, the problem becomes a standard empirical risk minimization (similar to (1)),

which can be solved by standard packages for linear SVM or logistic regression. When $\boldsymbol{w}$ is fixed, we use stochastic gradient descent (SGD) to solve the following subproblem with respect to $V$:

$$\underset{V \in \mathbb{R}^{k \times n}}{\operatorname{argmin}} \sum_{t=1}^{T} \left( \ell(r_t + \sum_{j,q \in I_t^+} \boldsymbol{v}_j^T \boldsymbol{v}_q - \sum_{j,q \in I_t^-} \boldsymbol{v}_j^T \boldsymbol{v}_q, o_t) + \sum_{j \in I_t^+ \cup I_t^-} \frac{\lambda_V}{d_j} \|\boldsymbol{v}_j\|_2^2 \right), \qquad (5)$$

where $d_j = |\{t : j \in I_t^+ \cup I_t^-\}|$ is number of games involving player $j$ and $r_t = \boldsymbol{w}^T(\boldsymbol{e}_t^+ - \boldsymbol{e}_t^-)$. The SGD update is then

$$\boldsymbol{v}_j \leftarrow \boldsymbol{v}_j - 2\eta(\ell'(s_t^+ - s_t^-)(\sum_{q \in I_t^+} \boldsymbol{v}_q) + (\lambda_V/d_j)\boldsymbol{v}_j) \text{ if } j \in I_t^+$$

$$\boldsymbol{v}_j \leftarrow \boldsymbol{v}_j - 2\eta(-\ell'(s_t^+ - s_t^-)(\sum_{q \in I_t^-} \boldsymbol{v}_q) + (\lambda_V/d_j)\boldsymbol{v}_j) \text{ if } j \in I_t^-.$$

Each SGD step only costs $O(kL)$ time by pre-computing $\sum_{q \in I_t^+} \boldsymbol{v}_q$ and $\sum_{q \in I_t^-} \boldsymbol{v}_q$, so Factorization HOI can scale to very large datasets.

### 4.2 Sample Complexity Analysis. How many games do we need?

To derive the theoretical guarantee, we first re-formulate (4). In this model, we can rewrite

$$s_t^+ = \boldsymbol{w}^T \boldsymbol{e}_t^+ + (\boldsymbol{e}_t^+)^T (V^T V) \boldsymbol{e}_t^+.$$

Therefore, by assuming $M = V^T V$, and using the fact that $\|M\|_* = \min_{V:M=V^T V} \|V\|_F^2$, the Factorization HOI (4) can be converted to the following nuclear norm regularization problem:

$$\min_{\boldsymbol{w}, M} \sum_{t=1}^{T} \ell(f_{\boldsymbol{w}, M}(\boldsymbol{e}_t^+, \boldsymbol{e}_t^-), o_t) + (\lambda_w/2)\|\boldsymbol{w}\|_2^2 + \lambda_V \|M\|_*,$$

where $f_{\boldsymbol{w}, M}(\boldsymbol{e}^+, \boldsymbol{e}^-) := \boldsymbol{w}^T(\boldsymbol{e}^+ - \boldsymbol{e}^-) + (\boldsymbol{e}^+)^T M \boldsymbol{e}^+ - (\boldsymbol{e}^-)^T M \boldsymbol{e}^-$. We then derive the guarantee for the following equivalent hard-constraint form:

$$\min_{\boldsymbol{w}, M} \frac{1}{T} \sum_{t \in \Omega} \ell(f_{\boldsymbol{w}, M}(\boldsymbol{e}_t^+, \boldsymbol{e}_t^-), o_t) \text{ s.t. } \|\boldsymbol{w}\|_2 \leq w, \ \|M\|_* \leq \mathcal{M}, \qquad (6)$$

where $\Omega$ is the set of observed group comparisons (there can be repeated pairs in $\Omega$). Assume $\boldsymbol{e}_t^+, \boldsymbol{e}_t^-$ are sampled from $\mathcal{E}$ defined by all the $n$-dimensional 0/1 vectors with $L$ ones, where $L$ is the number of players on each team. Both of them are sampled from a fixed distribution, under the sampling with replacement model. Our goal is to bound the expected error defined by

$$R(f) := \mathbb{E}\left[ \mathbb{1}\big( \operatorname{sgn}(f(\boldsymbol{e}_t^+, \boldsymbol{e}_t^-)) \neq \operatorname{sgn}(o_t) \big) \right].$$

More specifically, we want to study the sample complexity of our model: how many samples do we need for our model to achieve small prediction error? We will show that the number of samples is proportional to the nuclear norm ($\mathcal{M}$) and the two norm ($w$) of the underlying solution, which can be small in many realistic scenarios. The sample complexity analysis is based on problem (6), but solving it is slow (due to the need of SVD). In practice, we solve problem (4) for large-scale problem. Note that this is a generalized low-rank model, so based on [9], solving (4) with gradient descent could converge to global minimum under certain assumptions. All the detailed proofs are included in the appendix.

We will need the notation of expected and empirical $\ell$-risk:

$$\text{Expected } \ell\text{-risk: } R_\ell(f) = \mathbb{E}[\ell(f(\boldsymbol{e}_t^+, \boldsymbol{e}_t^-), o_t)]$$

$$\text{Empirical } \ell\text{-risk: } \hat{R}_\ell(f) = \frac{1}{T} \sum_{t=1}^{T} \ell(f(\boldsymbol{e}_t^+, \boldsymbol{e}_t^-), o_t).$$

Let the set of feasible $\boldsymbol{w}, M$ defined as $\Theta = \{(\boldsymbol{w}, M) | \|\boldsymbol{w}\|_2 \leq w$ and $\|M\|_* \leq \mathcal{M}\}$ and $\mathcal{F}_\Theta = \{f_{\boldsymbol{w}, M} \mid (\boldsymbol{w}, M) \in \Theta\}$. We then have the following lemma:

**Lemma 1.** *Let $\ell$ be a loss function with Lipschitz constant $L_\ell$ bounded by $\mathcal{B}$ with respect to its first argument, and $\delta$ be a constant where $0 < \delta < 1$. Then with probability at least $1 - \delta$, the expected $\ell$-risk is upper bounded by:*

$$R_\ell(f) \leq \hat{R}_\ell(f) + \min\left\{ 4w\sqrt{\frac{L}{T}} + 8L_\ell \mathcal{M}L\sqrt{\frac{\log(2n)}{T}}, \sqrt{\frac{144c_3 L_\ell \mathcal{B}\sqrt{L}(w + \sqrt{nL}\mathcal{M})\sqrt{N}}{T}} \right\} +$$

$$\mathcal{B}\sqrt{\frac{\log\frac{1}{\delta}}{2T}},$$

*for all $f \in \mathcal{F}_\Theta$, where $T$ is number of games and $c_3$ is a universal constant. For other constants please see Section 2 for details.*

Lemma 1 states that the expected loss will be close to empirical loss if $w$ and $\mathcal{M}$ are small, and the bound is proportional to $w, \mathcal{M}$ and inverse proportional to $\sqrt{T}$.

Now we discuss the recovery guarantee if the score $s_t^+, s_t^-$ are generated from some underlying model following $s_t^+ = \sum_{j \in I_t^+} w_j + \sum_{j \in I_t^+} \sum_{q \in I_t^+} M_{jq}$ with $\|w\| \leq w$ and $\|M\|_* \leq \mathcal{M}$, and assume the assumptions in Lemma 1 are also satisfied. We then have the following two theorems:

**Theorem 1.** *(Guarantee for score difference case). Let $\delta \in (0,1)$ be a constant. Suppose the following assumptions hold:*

- *$T$ clean comparisons[1] $o_t = s_t^+ - s_t^-$ are observed.*
- *The convex surrogate loss functions $\ell$ is bounded for each $o_t$, with $\ell(z,z) = 0$.*

*with probability at least $1 - \delta$, the optimal $f^*$ from problem (6) satisfies:*

$$R(f^*) \leq \min\left\{ O\left(\frac{w}{\sqrt{T}} + \mathcal{M}\sqrt{\frac{\log(2n)}{T}}\right), O\left(\sqrt{\frac{(w + \sqrt{nL}\mathcal{M})\sqrt{N}}{T}}\right) \right\} + O\left(\sqrt{\frac{\log\frac{1}{\delta}}{T}}\right),$$

When we can only observe the winning/losing game results ($o_t = \text{sgn}(s_t^+ - s_t^-)$), we have the following guarantee.

**Theorem 2.** *(Guarantee for binary result case). Let $\delta \in (0,1)$ be a constant. Suppose the following assumptions hold:*

- *$T$ clean comparisons $o_t = sgn(s_t^+ - s_t^-)$ are observed.*
- *The convex surrogate loss functions $\ell$ is bounded for each $o_t$.*

*With probability at least $1 - \delta$, the optimal $f^*$ from problem (6) satisfies:*

$$R(f^*) \leq O\left(\hat{R}_\ell(f^*) - R_\ell^*\right) + \min\left\{ O\left(\frac{w}{\sqrt{T}} + \mathcal{M}\sqrt{\frac{\log(2n)}{T}}\right),\right.$$

$$\left. O\left(\sqrt{\frac{(w + \sqrt{nL}\mathcal{M})\sqrt{N}}{T}}\right) \right\} + O\left(\sqrt{\frac{\log\frac{1}{\delta}}{T}}\right)$$

*where $R_\ell^* = \inf_f R_\ell(f)$.*

In Theorem 2, the term $\hat{R}_\ell(f^*) - R_\ell^*$ may not be zero but will be small, depending on how we define loss. In summary, after observing $T$ samples, the expected error will be $O(\min(w + \mathcal{M}, (w + \mathcal{M})^{1/2}N^{1/4}))/\sqrt{T})$ in Theorem 1. The second term has less dependency on $w$ and $\mathcal{M}$, but will be large for large $L$ (number of players per team), since $N = O(n^L)$. However, we take the minimum for these two terms, so in either cases the sample complexity will be small if the nuclear norm $\mathcal{M}$ and two norm of $w$ are small. We have the same conclusion for binary $(+1/-1)$ observations when $\hat{R}_\ell(f^*) - R_\ell^* = O(\epsilon)$.

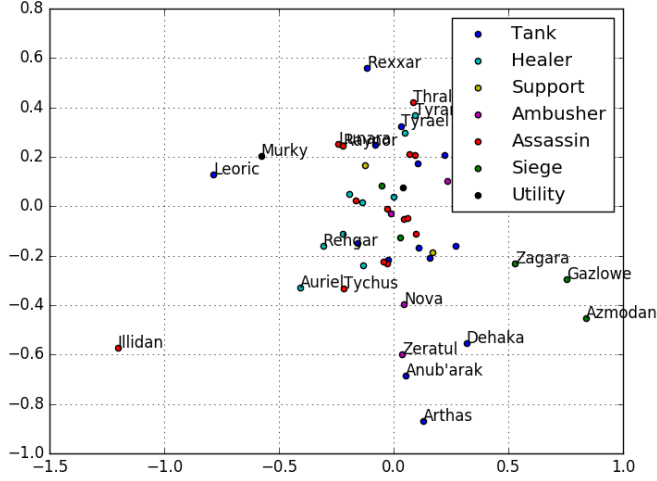

Figure 1: Projection of interaction features for each hero ($v_i$) to 2-D space. Colors represents the official categorization for these heroes. This low-dimensional representation reveals some interesting patterns for pairwise interactions between heroes in Heroes of the Storm.

All the previous discussion is based on the assumption that we can observe clean comparisons. However, in practice, we usually observe noisy comparisons. We use a standard "flip sign model"[19], where each comparison result is independently flipped ($\tilde{o}_t = -\text{sgn}(o_t)$) with probability $\rho_f \in [0, 0.5)$, where $\tilde{o}_t$ is the observed flipped result. The following theorem shows that with noisy comparisons, we just need slightly more samples, depending on the noise level.

**Theorem 3.** (Guarantee for noisy comparisons). *Let each $o_t$ is now observed under the "flip sign model" with $\rho_f \in [0, 0.5)$. Then by solving Factorization HOI with squared loss,*

$$\min\left\{O\left(\frac{1}{1-2\rho_f}\left(\frac{w}{\sqrt{T}} + \mathcal{M}\sqrt{\frac{\log(2n)}{T}}\right)\right), O\left(\sqrt{\frac{(w+\sqrt{nL}\mathcal{M})\sqrt{N}}{(1-\rho_f)T}}\right)\right\} + O\left(\sqrt{\frac{\log(\frac{1}{\delta})}{T}}\right)$$

*comparisons suffice to guarantee an $\epsilon$-accurate result.*

Theorem 3 demonstrates that in noisy comparison case, Factorization HOI can achieve $\epsilon-$accurate result with the same order of sample complexity as in clean comparison cases, but with a extra price, which is a $\frac{1}{1-\rho_f}$ or $\frac{1}{\sqrt{1-\rho_f}}$ factor.

## 5 Experimental Results

We include the following algorithms in our experiments:

- Basic HOI: the proposed basic model using pairwise information with squared hinge loss (see eq (2)).
- Factorization HOI: the proposed model in eq (4) with squared hinge loss, which approximates the pairwise interaction by a factor form.
- Trueskill [12]: the state-of-the-art algorithm used in all the online game matching engines. Since it is an online algorithm, we test the performance after running 1 epoch and 10 epochs. We do not observe any accuracy gain after 10 epochs.
- Bradley-Terry model [13]: the generalized Bradley-Terry model for group comparison data.
- Logistic Regression (LR): another baseline for individual score model (1) using logistic loss. We have not seen this algorithm in the literature, but we found this simple approach works quite well so we include it into comparison.

**Datasets and parameter settings.** We consider datasets from two online games: Heroes of the Storm (HotS) and Dota 2. Both of them are Multiplayer Online Battle Arena (MOBA) games. In each game, two five-player teams fight with each other on a map. Each player can choose one of the

heroes (characters), and each hero has different abilities. There are totally around 60 heroes in HotS and 100 heroes in Dota 2. For each dataset we consider two tasks: (1) we consider each hero as an individual so that each game we get the group comparisons between 5 heroes versus another 5 heroes. And the goal is to predict the outcome of the games. Since there are only around 100 heroes, the parameter space will not be too large even for learning $n^2$ pairwise interactions. (2) We also consider each player as an individual, so that each group comparison is between 5 players versus another 5 players. In this case, there can be tens of thousands of players, so the parameter space is huge.

We collect the following three sets of data. For HotS tournament matches, we download all matching records provided by Hotslog[2] for the years of 2015 and 2016. For HotS public game data, we crawl the matching history of Master players in Hotslog. There are three game modes for public games—quick match, team league, and hero league. Here we only consider the hero league data since it is closer to the official tournament games. For Dota 2, we download the recent data from OpenDota [3]. We focus on a set of "notable players" (defined by the website), and get all their matching data in public games. For each dataset, we have two different views, taking heroes as individuals ($n$) or taking players as individuals ($n$). So we have 6 datasets in total, as listed in Table 1.

For each dataset, we randomly divided the games into 80% for training and 20% for testing. For all the methods, we cross validate on the training set to choose the best parameter, and then use the best parameter to train a final model, which is then evaluated on the testing set. For our model, determining the values of $k$ is a trade-off between the model efficiency and accuracy. In our experiments, we choose $k$ by cross validation. Accuracy is evaluated by number of correct predicted games divided by the total number of testing games. The results are presented in Table 2. Note that Basic HOI will generate $n^2$ parameters, so it runs out of memory for some datasets. We have the following findings:

- Our proposed algorithms, Basic HOI and Factorization HOI are always better than individual models, which indicates that higher order information is useful for modeling group comparisons. Moreover, we observe that higher order information is particularly useful for tournament data (HotS tournament), which makes sense because tournament players are more advanced and have better teamwork. The outcome of a professional game is often determined by some good use of "combo".[4]
- For hero data, since the number of individuals is small, Basic HOI is able to learn a good model for all the individual scores and thus slightly outperforms Factorization HOI. However, when the number of individuals grow to thousands (e.g., two HotS player datasets), Basic HOI has too many parameters to learn and suffers from over-fitting, so the accuracy is lower than Factorization HOI. Furthermore, Factorization HOI is able to scale to large amount of individuals (e.g., 30,000 players in Dota 2), while Basic HOI will run out of memory since it requires $O(n^2)$ memory.

Finally, in addition to better prediction accuracy, our model reveals interesting patterns that cannot be discovered by individual scores. First, we extract the top-5 and bottom-5 hero pairs for HotS Tournament data (see Table 3). Among them, one of the top-5 pairs, (Reghar, Illidan), is a famous strong combination recognized by professional players, while most of the bottom-5 pairs are clearly not good since they are heroes with repeated functions. Our results can thus guide the players and professional coaches for selecting heroes. For example, Illidan works well with Reghar, but is very bad with Thrall. We also extract the top-5 and bottom-5 pairs based on Bradley-Terry and Trueskill (see Table 4). It is obvious that the top-5/bottom-5 pairs based on Bradley-Terry and Trueskill are totally different from pairs got from our method, which shows that our method can capture interaction effect that are not explored well in the previous methods. In addition, we project the learned latent factors $v_i$ in the Factorization HOI model to a 2D space by PCA in Figure 1. These vectors characterize the pairwise interaction between heroes. We observe many interesting patterns. For example, most of the siege heroes are on the right bottom area of the space; Murky and Leoric are on the top-left corner, where they have similar behavior (this is actually an important combination that helped C9 team to win the 2015 Heroes of the Storm championship). Illidan is in the very left-botton corner, which means it is very good with other heroes in the third quadrant, but very bad with the heroes in the first quadrant.

Table 1: Dataset Statistics

| Datasets | HotS Tournament (Hero) | HotS Tournament (Player) | HotS Public (Hero) | HotS Public (Player) | Dota 2 (Hero) | Dota 2 (Player) |
|---|---|---|---|---|---|---|
| Number of Games ($T$) | 9,610 | 9,610 | 139,462 | 139,462 | 46,459 | 46,459 |
| Number of Individuals ($n$) | 54 | 3,470 | 62 | 7,251 | 113 | 30,452 |

Table 2: Performance of the proposed algorithm and other algorithms. The numbers are prediction accuracy (%), and "oom" indicates out of memory here.[5]

| Datasets | LR | Trueskill (1) | Trueskill (10) | Bradley-Terry | Basic HOI | Factorization HOI |
|---|---|---|---|---|---|---|
| HotS Tournament (H) | 59.73 | 62.90 | 58.48 | 59.52 | **80.59** | 77.84 |
| HotS Tournament (P) | 83.45 | 80.02 | 84.50 | 84.18 | 83.89 | **85.17** |
| HotS Public (H) | 54.34 | 53.36 | 53.06 | 53.50 | 54.45 | **54.75** |
| HotS Public (P) | 54.01 | 53.64 | 53.87 | 53.92 | 53.39 | **55.76** |
| Dota 2 (H) | 61.64 | 52.50 | 52.61 | 61.37 | **65.34** | 63.72 |
| Dota 2 (P) | 65.98 | 62.16 | 64.26 | 62.72 | oom | **68.25** |

Table 3: Top-5 pairs and bottom-5 hero pairs learned by our model on Heroes of the storm tournament data.

| Top 5 pairs | Bottom 5 pairs |
|---|---|
| (Lunara, Leoric) | (Raynor, Zeratul) |
| (Kerrigen, Sylvanas) | (Illidan, Thrall) |
| (Reghar, Illidan) | (Sonya, Zeratul) |
| (Chen, Jaina) | (Muradin, Lt. Morales) |
| (Thrall, Valla) | (Anub'arak, Illidan) |

Table 4: Top-5 and bottom-5 pairs for Trueskill and Bradley-Terry Method

| Top 5 pairs (Trueskill) | Bottom 5 pairs (Trueskill) | Top 5 pairs (BTL) | Bottom 5 pairs (BTL) |
|---|---|---|---|
| (Auriel,Kerrigan) | (Chromie,Sgt.Hammer) | (Auriel,Medivh) | (Chromie,Sgt.Hammer) |
| (Auriel,Tracer) | (Chromie,The Butcher) | (Auriel,The Lost Vikings) | (Chromie,Gazlowe) |
| (Auriel,Rexxar) | (Chromie,Valla) | (Auriel,Rehgar) | (Chromie,The Butcher) |
| (Auriel,The Lost Vikings) | (Chromie,Gazlowe) | (Auriel,Kerrigan) | (Chromie,Tychus) |
| (Auriel,Kerrigan) | (Chromie,Tychus) | (Auriel,Brightwing) | (Chromie,Artanis) |

# 6 Conclusions

Previous models for group comparisons are all based on individual score models. In this paper, we develop novel algorithms to utilize higher order interactions between players. The proposed algorithm achieves much higher accuracy than existing methods, indicating that modeling higher order interaction is crucial for mining group comparison data.

# 7 Acknowledgement

The paper is partially supported by the support of NSF via IIS-1719097, Intel Faculty Award, Google Cloud and Nvidia.

## Footnotes

[1]"clean comparison" means that the observed outcomes are noiseless.

[2] https://www.hotslogs.com/Default

[3] https://www.opendota.com

[4] In games, a "combo" indicates a set of actions performed in sequence that yield a significant benefit or advantage. A "combo" usually requires very precise timing, so is more commonly used by advanced players.

[5]In online games, "oom" often stands for out-of-mana.

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
