[Supplementary Material]

# 8 Appendix: Proofs

For the convenience of notation, in the following part we will use $\boldsymbol{x}_m$ and $\boldsymbol{x}_l$ to represent $\boldsymbol{e}_t^+$ and $\boldsymbol{e}_t^-$, so $\boldsymbol{x}_m$ and $\boldsymbol{x}_l$ are the indicator vectors of team $m$ and team $l$ respectively. We use $o_{ml}$ to represent the game outcome between the two teams.

## 8.1 Proof of Lemma 1

We will use the following lemma to prove Lemma 1 (which can be found in [2]):

**Lemma 2.** *(Bound on Expected $\ell$-risk [2]). Let $\ell$ be a loss function with Lipschitz constant $L_\ell$ bounded by $\mathcal{B}$ with respect to its first argument, and $\delta$ be a constant where $0 < \delta < 1$. Let $\mathfrak{R}(F_\Theta)$ be the Rademacher complexity of the function class $F_\Theta$ (w.r.t $\Omega$ and associated with $\ell$) defined as:*

$$\mathfrak{R}(F_\Theta) = \mathbb{E}_\sigma[\sup_{f \in F_\Theta} \frac{1}{T} \sum_{\alpha=1}^{T} \sigma_\alpha \ell(f(\boldsymbol{x}_{m_\alpha}, \boldsymbol{e}_{l_\alpha}), o_{m_\alpha l_\alpha})], \tag{7}$$

*where each $\sigma_\alpha$ takes values $\{\pm 1\}$ with equal probability. Then with probability at least $1 - \delta$, for all $f \in F_\Theta$ we have:*

$$R_\ell(f) \leq \hat{R}_\ell(f) + 2\mathbb{E}_\Omega[\mathfrak{R}(F_\Theta)] + \mathcal{B}\sqrt{\frac{\log \frac{1}{\delta}}{2T}}.$$

**Proof (of Lemma 1):** Note that $\boldsymbol{x}_m$ is indicator vector of team $m$, thus $\sqrt{L} = \max_{m \in [N]} \|\boldsymbol{x}_m\|_2$, where $L$ is the number of players on each team. The model complexity of (7) can be bounded by:

$$\mathfrak{R}(F_\Theta) = \mathbb{E}_\sigma[\sup_{f \in F_\Theta} \frac{1}{T} \sum_{\alpha=1}^{T} \sigma_\alpha \ell(f_{\boldsymbol{w},M}(\boldsymbol{x}_{m_\alpha}, \boldsymbol{x}_{l_\alpha}), o_{m_\alpha l_\alpha})]$$

$$\leq L_\ell \mathbb{E}_\sigma[\sup_{f \in F_\Theta} \frac{1}{T} \sum_{\alpha=1}^{T} \sigma_\alpha f_{\boldsymbol{w},M}(\boldsymbol{x}_{m_\alpha}, \boldsymbol{x}_{l_\alpha})]$$

$$= \frac{L_\ell}{T} \mathbb{E}_\sigma[\sup_{\|\boldsymbol{w}\|_2 \leq w, \|M\|_* \leq \mathcal{M}} \sum_{\alpha=1}^{T} \sigma_\alpha(\boldsymbol{w}^T(\boldsymbol{x}_{m_\alpha} - \boldsymbol{x}_{l_\alpha})$$
$$+ \boldsymbol{x}_{m_\alpha}^T M \boldsymbol{x}_{m_\alpha} - \boldsymbol{x}_{l_\alpha}^T M \boldsymbol{x}_{l_\alpha})]$$

$$\leq L_\ell \mathbb{E}_\sigma[\sup_{\|\boldsymbol{w}\|_2 \leq w} \frac{1}{T} \sum_{\alpha=1}^{T} \sigma_\alpha \boldsymbol{w}^T(\boldsymbol{x}_{m_\alpha} - \boldsymbol{x}_{l_\alpha})$$

$$+ \sup_{M:\|M\|_* \leq \mathcal{M}} \frac{1}{T} \sum_{\alpha=1}^{T} \sigma_\alpha \text{trace}(M\boldsymbol{x}_{m_\alpha} \boldsymbol{x}_{m_\alpha}^T)$$

$$+ \sup_{M:\|M\|_* \leq \mathcal{M}} \frac{1}{T} \sum_{\alpha=1}^{T} \sigma_\alpha \text{trace}(M\boldsymbol{x}_{l_\alpha} \boldsymbol{x}_{l_\alpha}^T)]$$

$$\leq 2w\sqrt{\frac{L}{T}} + 4L_\ell \mathcal{M} L \sqrt{\frac{\log(2n)}{T}}$$

This is the first bound of $\mathfrak{R}(F_\Theta)$. Note that the last inequality comes from the condition $\sqrt{L} = \max_{m \in [N]} \|\boldsymbol{x}_m\|_2$ and Theorem 1 in [14]. In the following part we derive another bound for $\mathfrak{R}(F_\Theta)$ that has better dependency to $w$ and $\mathcal{M}$. We can rewrite $\mathfrak{R}(F_\Theta)$ as:

$$\mathfrak{R}(F_\Theta) = \mathbb{E}_\sigma[\sup_{f \in F_\Theta} \frac{1}{T} \sum_{\alpha=1}^{T} \sigma_\alpha \ell(f_{\boldsymbol{w},M}(\boldsymbol{x}_{m_\alpha}, \boldsymbol{x}_{l_\alpha}), o_{m_\alpha l_\alpha})]$$

$$= \mathbb{E}_\sigma[\sup_{f \in F_\Theta} \frac{1}{T} \sum_{(m,l)} \Gamma_{ml} \ell(f_{\boldsymbol{w},M}(\boldsymbol{x}_m, \boldsymbol{x}_l), o_{ml})]$$

where $\Gamma \in \mathbb{R}^{N \times N}$ with each entry defined as $\Gamma_{ml} = \sum_{\alpha:m_\alpha=m,l_\alpha=l} \sigma_\alpha$. Use the same method in [18], we can decompose $\Gamma$ into two matrices $A$ and $B$, where $A$ contains the "heavily-hit" entries, and $B$ the "lightly-hit" entries, where the two types of entries are differentiated according to some threshold $p$.

Given $m, l$, let $h_{m,l} = |\{\alpha : m_\alpha = m, l_\alpha = l\}|$ be the number of times the sample $\Omega$ hits entry $m, l$. Let $p > 0$ be an arbitrary parameter to be specified later, and define

$$A_{m,l} = \begin{cases} \Gamma_{m,l} & h_{m,l} > p \\ 0 & h_{m,l} \leq p \end{cases} \quad B_{m,l} = \begin{cases} 0 & h_{m,l} > p \\ \Gamma_{m,l} & h_{m,l} \leq p \end{cases}$$

Clearly, $\Gamma = A + B$. We can write $\mathfrak{R}(F_\Theta)$ as:

$$\mathfrak{R}(F_\Theta) = \mathbb{E}_\sigma \Big[ \sup_{f \in F_\Theta} \frac{1}{T} \sum_{(m,l)} A_{ml} \ell(f_{\boldsymbol{w},M}(\boldsymbol{x}_m, \boldsymbol{x}_l), o_{ml}) \Big]$$

$$+ \mathbb{E}_\sigma \Big[ \sup_{f \in F_\Theta} \frac{1}{T} \sum_{(m,l)} B_{ml} \ell(f_{\boldsymbol{w},M}(\boldsymbol{x}_m, \boldsymbol{x}_l), o_{ml}) \Big] \tag{8}$$

Since $|\ell(f_{\boldsymbol{w},M}(\boldsymbol{x}_m, \boldsymbol{x}_l), o_{ml})| \leq \mathcal{B}$, the first term of (8) can be upper bounded by

$$\frac{1}{T} \mathbb{E}_\sigma \Big[ \mathcal{B} \sum_{(m,l)} A_{ml} \Big] \leq \frac{\mathcal{B}}{\sqrt{p}}$$

Using the Rademacher contraction principle, the second term of (8) can be upper bounded by:

$$\frac{L_\ell}{T} \mathbb{E}_\sigma \Big[ \sup_{\|\boldsymbol{w}\|_2 \leq w, \|M\|_* \leq \mathcal{M}} \sum_{(m,l)} B_{ml} f_{\boldsymbol{w},M}(\boldsymbol{x}_m, \boldsymbol{x}_l) \Big]$$

$$= \frac{L_\ell}{T} \mathbb{E}_\sigma \Big[ \sup_{\|\boldsymbol{w}\|_2 \leq w, \|M\|_* \leq \mathcal{M}} \sum_{(m,l)} B_{ml} (\boldsymbol{w}^T(\boldsymbol{x}_m - \boldsymbol{x}_l) + \boldsymbol{x}_m^T M \boldsymbol{x}_m - \boldsymbol{x}_l^T M \boldsymbol{x}_l) \Big]$$

$$\leq \frac{L_\ell}{T} \mathbb{E}_\sigma \Big[ \sup_{\|\boldsymbol{w}\|_2 \leq w} \sum_{(m,l)} B_{ml} \boldsymbol{w}^T(\boldsymbol{x}_m - \boldsymbol{x}_l) \Big] + \frac{L_\ell}{T} \mathbb{E}_\sigma \Big[ \sup_{\|M\|_* \leq \mathcal{M}} \sum_{(m,l)} B_{ml} \boldsymbol{x}_m^T M \boldsymbol{x}_m \Big]$$

$$+ \frac{L_\ell}{T} \mathbb{E}_\sigma \Big[ \sup_{\|M\|_* \leq \mathcal{M}} \sum_{(m,l)} B_{ml} \boldsymbol{x}_l^T M \boldsymbol{x}_l \Big]$$

$$= \frac{L_\ell}{T} \mathbb{E}_\sigma \sup_{\|\boldsymbol{w}\|_2 \leq w} \sum_{(m,l)} B_{ml} \boldsymbol{w}^T(\boldsymbol{x}_m - \boldsymbol{x}_l) + \frac{L_\ell}{T} \mathbb{E}_\sigma \Big[ \sup_{\|M\|_* \leq \mathcal{M}} \sum_{(m,l)} B_{ml} (XMX^T)_{mm} \Big]$$

$$+ \frac{L_\ell}{T} \mathbb{E}_\sigma \Big[ \sup_{\|M\|_* \leq \mathcal{M}} \sum_{(m,l)} B_{ml} (XMX^T)_{ll} \Big]$$

$$\leq \frac{2w L_\ell \sqrt{L}}{T} \mathbb{E}_\sigma [\|B\|_2] + \frac{2L_\ell}{T} \mathbb{E}_\sigma \Big[ \sup_{W:\|W\|_* \leq \mathcal{W}} \|B\|_2 \|XX^T\|_* \Big]$$

$$\leq \frac{2w L_\ell \sqrt{L}}{T} \mathbb{E}_\sigma [\|B\|_2] + \frac{2L_\ell \mathcal{M} L \sqrt{n}}{T} \mathbb{E}_\sigma [\|B\|_2]$$

$$\leq \frac{8.8 c_3 L_\ell \sqrt{L}(w + \sqrt{nL}\mathcal{M})\sqrt{p}\sqrt{N}}{T}.$$

Choosing $p = \frac{\mathcal{B}T}{8.8 c_3 L_\ell \sqrt{L}(w+\sqrt{nL}\mathcal{M})\sqrt{N}}$, (8) is bounded by

$$\sqrt{\frac{36 c_3 L_\ell \mathcal{B} \sqrt{L}(w + \sqrt{nL}\mathcal{M})\sqrt{N}}{T}}.$$

Applying Lemma 2, we can get the bound in Lemma 1.

## 8.2 Proof of Theorem 1

**Lemma 3.** *(Consistency of Excess Risk [1]). Let $\ell$ be a convex surrogate loss function. Then there exists a strictly increasing function $\Psi$, $\Psi(0) = 0$, such that for all measurable $f$:*

$$R(f) - R^* \leq \Psi(R_\ell(f) - R_\ell^*),$$

*where $R^* = \inf_f R(f)$ and $R_\ell^* = \inf_f R_\ell(f)$.*

**Proof (of Theorem 1).** When we can observe the score differences, $o_{ml} = s_t^+ - s_t^-$. Let $f_{\boldsymbol{w},M}^*(\boldsymbol{x}_m, \boldsymbol{x}_l), \theta^* = [\boldsymbol{w}^*; M^*] \in \Theta$ to be the optimal solution of problem (6). If the scores $s_t^+, s_t^-$ are generated from some underlying model $w^*, M^*$ following $s_t^+ = \sum_{j \in I_t^+} w_j^* + \sum_{j \in I_t^+} \sum_{q \in I_t^+} M_{jq}^*$ with $\|\boldsymbol{w}^*\| \leq w$ and $\|M^*\|_* \leq \mathcal{M}$, we have $\ell(\boldsymbol{w}^{T^*}(\boldsymbol{x}_m - \boldsymbol{x}_l) + \boldsymbol{x}_m^T M^* \boldsymbol{x}_m - \boldsymbol{x}_l^T M^* \boldsymbol{x}_l, s_t^+ - s_t^-) = \ell(s_t^+ - s_t^-, s_t^+ - s_t^-) = 0$. Thus, we can get $\hat{R}(f^*)$. Apparently, $R^* = R_\ell^* = 0$, so Lemma (3) here is:

$$R(f^*) \leq \Psi(R_\ell(f^*)),$$

Therefore, applying Lemma 1 we can get:

$$R_\ell(f^*) \leq \min\left\{4w\sqrt{\frac{L}{T}} + 8L_\ell \mathcal{M}L\sqrt{\frac{\log(2n)}{T}}, \sqrt{\frac{144c_3 L_\ell \mathcal{B}\sqrt{L}(w + \sqrt{nL}\mathcal{M})\sqrt{N}}{T}}\right\} + \mathcal{B}\sqrt{\frac{\log\frac{1}{\delta}}{2T}}$$

Let $L_\Psi$ be the bounded Lipschitz constant for $\Psi$. Then we can derive:

$$R(f^*) \leq \Psi(R_\ell(f^*))$$
$$\leq L_\Psi \left( \min\left\{ 4w\sqrt{\frac{L}{T}} + 8L_\ell \mathcal{M}L\sqrt{\frac{\log(2n)}{T}}, \right.\right.$$
$$\left. \sqrt{\frac{144c_3 L_\ell \mathcal{B}\sqrt{L}(w + \sqrt{nL}\mathcal{M})\sqrt{N}}{T}}\right\} + \mathcal{B}\sqrt{\frac{\log\frac{1}{\delta}}{2T}} \right)$$
$$= \min\left\{ O\left(\frac{w}{\sqrt{T}} + \mathcal{M}\sqrt{\frac{\log(2n)}{T}}\right), O\left(\sqrt{\frac{(w + \sqrt{nL}\mathcal{M})\sqrt{N}}{T}}\right)\right\}$$
$$+ O\left(\sqrt{\frac{\log\frac{1}{\delta}}{T}}\right)$$

## 8.3 Proof of Theorem 2

When we can only observe the winning/losing game results, $o_t = \text{sgn}(s_t^+ - s_t^-)$. $R^* = 0$ still holds, but $R_\ell^*$ may not be zero. Applying Lemma (3), we have:

$$R(f^*) \leq \Psi(R_\ell(f^*) - R_\ell^*).$$

Using Lemma 1, we can bound $R_\ell(f^*) - R_\ell^*$ by:

$$R_\ell(f^*) - R_\ell^* \leq \hat{R}_\ell(f^*) - R_\ell^* + \min\left\{4w\sqrt{\frac{L}{T}} + 8L_\ell \mathcal{M}L\sqrt{\frac{\log(2n)}{T}}, \right.$$
$$\left. \sqrt{\frac{144c_3 L_\ell \mathcal{B}\sqrt{L}(w + \sqrt{nL}\mathcal{M})\sqrt{N}}{T}}\right\} + \mathcal{B}\sqrt{\frac{\log\frac{1}{\delta}}{2T}}$$

Therefore, we can derive:

$$R(f^*) \leq \Psi(R_\ell(f^*) - R_\ell^*)$$

$$\leq L_\Psi \left( \hat{R}_\ell(f^*) - R_\ell^* + \min \left\{ 4w\sqrt{\frac{L}{T}} + 8L_\ell\mathcal{M}L\sqrt{\frac{\log(2n)}{T}}, \right.\right.$$

$$\left.\left. \sqrt{\frac{144c_3 L_\ell \mathcal{B}\sqrt{L}(w + \sqrt{n\bar{L}}\mathcal{M})\sqrt{N}}{T}} \right\} + \mathcal{B}\sqrt{\frac{\log\frac{1}{\delta}}{2T}} \right)$$

$$= O\left( \hat{R}_\ell(f^*) - R_\ell^* \right) + \min \left\{ O\left( \frac{w}{\sqrt{T}} + \mathcal{M}\sqrt{\frac{\log(2n)}{T}} \right), \right.$$

$$\left. O\left( \sqrt{\frac{(w + \sqrt{n\bar{L}}\mathcal{M})\sqrt{N}}{T}} \right) \right\} + O\left( \sqrt{\frac{\log\frac{1}{\delta}}{T}} \right)$$

## 8.4   Proof of Theorem 3

Theorem 3 follows directly from the following theorem provided that $\min_{f \in \mathcal{F}_\Theta} R_\ell - R_\ell^* = O(\epsilon)$, so prove Theorem 4 will suffice.

**Theorem 4.** (Kendall's Tau guarantee for noisy comparisons from flip sign model). *Let $\delta$ be any constant such that $0 < \delta < 1$. Suppose that we observe $T$ noisy group comparisons under the flip sign model parameterized by some noise level $0 \leq \rho_f \leq 0.5$.*

Consider the following problem:

$$\min_{\boldsymbol{w},M} \frac{1}{T} \sum_{(m,l)\in\Omega} [\boldsymbol{w}^T(\boldsymbol{x}_m - \boldsymbol{x}_l) + \boldsymbol{x}_m^T M\boldsymbol{x}_m - \boldsymbol{x}_l^T M\boldsymbol{x}_l - \tilde{o}_{ml}]^2, \tag{9}$$

$$\text{s.t. } \|\boldsymbol{w}\|_2 \leq (1 - 2\rho_f)w, \ \ \|M\|_* \leq (1 - 2\rho_f)\mathcal{M}, \ \ \tilde{o}_{ml} \sim D_{\rho_f}$$

where the distribution $D_{\rho_f}$ is defined by:

$$P(\tilde{o}_{ml} = +1 | sgn(o_{ml}) = -1)$$
$$= P(\tilde{o}_{ml} = -1 | sgn(o_{ml}) = +1)$$
$$= \rho_f$$

where $o_{ml}$ represents the clean comparison result. Then with probability at least $1 - \delta$, the optimal $f^*$ of the problem satisfies:

$$R(f^*) \leq O\left( \min_{f \in \mathcal{F}_\Theta} R_\ell - R_\ell^* \right) + \min \left\{ O\left( \frac{1}{1 - 2\rho_f}(\frac{w}{\sqrt{T}} + \mathcal{M}\sqrt{\frac{\log(2n)}{T}}) \right), \right.$$

$$\left. O\left( \sqrt{\frac{(w + \sqrt{n\bar{L}}\mathcal{M})\sqrt{N}}{(1 - 2\rho_f)T}} \right) \right\} + O\left( \sqrt{\frac{\log\frac{1}{\delta}}{T}} \right)$$

To prove Theorem 4 we need the following lemma. We will give the proof of Theorem 4 after Lemma 4.

**Lemma 4.** (Equivalence of Problem (9) with Unbiased Estimator). *The problem (9) is equivalent to the following optimization problem:*

$$\min_{\tilde{\boldsymbol{w}},\tilde{M}} \frac{1}{T} \sum_{(m,l)\in\Omega} \tilde{\ell}(\tilde{\boldsymbol{w}}^T(\boldsymbol{x}_m - \boldsymbol{x}_l) + \boldsymbol{x}_m^T \tilde{M}\boldsymbol{x}_m - \boldsymbol{x}_l^T \tilde{M}\boldsymbol{x}_l, \tilde{o}_{ml}) \tag{10}$$

$$\text{s.t. } \|\tilde{\boldsymbol{w}}\|_2 \leq w, \ \ \|\tilde{M}\|_* \leq \mathcal{M},$$

*where $\tilde{\ell}(t, y)$ is an unviased estimator of squared loss from noisy comparisons defined by:*

$$\tilde{\ell}(t, y) = \frac{(1 - \rho_f)(t - y)^2 - \rho_f(t + y)^2}{1 - 2\rho_f}$$

*Furthermore, the optimal solution of the problem ([10]), denoted as $\tilde{\theta}^*$, satisfies:*

$$\theta^* = (1 - 2\rho_f)\tilde{\theta}^* \tag{11}$$

*where $\theta^*$ is the optimal solution of the problem ([9]).*

With Lemma [4], we will give proof of Theorem [4] in the following part. Proof of Lemma [4] is provided at the end.

*Proof (of Theorem [4]).* Let $\tilde{\theta}^*/\tilde{f}^*$ denote the optimal parameter/function of problem ([10]). Then from Theorem 3 of [15], we can guarantee that with probability at least $1 - \delta$, the risk of $\tilde{f}^*$ w.r.t. clean distribution is bounded by:

$$R_\ell(\tilde{f}^*) \leq \min_{f \in \mathcal{F}_\Theta} R_\ell(f) + \frac{8L_\ell}{1 - 2\rho_f}\mathbb{E}_\Omega[\mathfrak{R}(F_\Theta)] + 2\sqrt{\frac{\log\frac{1}{\delta}}{2T}}. \tag{12}$$

From Lemma [4] we know that $\theta^* = (1-2\rho_f)\tilde{\theta}^*$, so the scores learned will be scaled by a $1-2\rho_f$ factor, but the relative scores and the comparison result will remain the same. This implies $R(f^*) = R(\tilde{f}^*)$. Finally, by applying Lemma [1] and Lemma [3] to ([12]), the claim of Theorem [4] can be obtained as:

$$R(\tilde{f}^*) \leq \Psi(R_\ell(\tilde{f}^*) - R_\ell^*)$$

$$\leq L_\Psi \left( \min_{f \in \mathcal{F}_\Theta} R_\ell(f) - R_\ell^* + \frac{8L_\ell}{1 - 2\rho_f} \min\left\{ 4w\sqrt{\frac{L}{T}} + \right.\right.$$

$$\left. 8L_\ell\mathcal{M}L\sqrt{\frac{\log(2n)}{T}}, \sqrt{\frac{144c_3L_\ell\mathcal{B}\sqrt{L}(w + \sqrt{n\bar{L}}\mathcal{M})\sqrt{N}}{T}} \right\} + \mathcal{B}\sqrt{\frac{\log\frac{1}{\delta}}{2T}} \right)$$

$$= O\left( \min_{f \in \mathcal{F}_\Theta} R_\ell - R_\ell^* \right) + \min\left\{ O\left( \frac{1}{1 - 2\rho_f}(\frac{w}{\sqrt{T}} + \mathcal{M}\sqrt{\frac{\log(2n)}{T}}) \right), \right.$$

$$\left. O\left( \sqrt{\frac{(w + \sqrt{n\bar{L}}\mathcal{M})\sqrt{N}}{(1 - 2\rho_f)T}} \right) \right\} + O\left( \sqrt{\frac{\log\frac{1}{\delta}}{T}} \right)$$

*Proof (of Lemma [4]).* First off, we rewrite the unbiased estimator of squared loss $\tilde{\ell}(t, y)$ as:

$$\tilde{\ell}(t, y) = t^2 - \frac{2t}{1 - 2\rho_f}y + y^2$$

$$= \left( t - \frac{y}{1 - 2\rho_f} \right)^2 + \left( y^2 - \frac{1}{1 - 2\rho_f}y^2 \right)$$

Therefore, problem ([10]) can be rewritten as:

$$\min_{\tilde{\boldsymbol{w}}, \tilde{M}} \frac{1}{T} \sum_{(m,l) \in \Omega} \tilde{\ell}(\tilde{\boldsymbol{w}}^T(\boldsymbol{x}_m - \boldsymbol{x}_l) + \boldsymbol{x}_m^T \tilde{M}\boldsymbol{x}_m - \boldsymbol{x}_l^T \tilde{M}\boldsymbol{x}_l, \tilde{o}_{ml})$$

$$\equiv \min_{\tilde{\boldsymbol{w}}, \tilde{M}} \frac{1}{T} \sum_{(m,l) \in \Omega} \left( \tilde{\boldsymbol{w}}^T(\boldsymbol{x}_m - \boldsymbol{x}_l) + \boldsymbol{x}_m^T \tilde{M}\boldsymbol{x}_m - \boldsymbol{x}_l^T \tilde{M}\boldsymbol{x}_l - \frac{\tilde{o}_{ml}}{1 - 2\rho_f} \right)^2 \tag{13}$$

$$\text{s.t. } \|\tilde{\boldsymbol{w}}\|_2 \leq w, \ \|\tilde{M}\|_* \leq \mathcal{M},$$

Now define two new variables as:

$$\boldsymbol{w} = (1 - 2\rho_f)\tilde{\boldsymbol{w}}$$
$$M = (1 - 2\rho_f)\tilde{M} \tag{14}$$

and substitute ([14]) to the problem ([13]). We can further derive an equivalent optimization problem w.r.t. $\boldsymbol{w}$ and $M$ as:

$$\min_{\boldsymbol{w}, M} \frac{1}{T} \sum_{(m,l) \in \Omega} \left( \boldsymbol{w}^T(\boldsymbol{x}_m - \boldsymbol{x}_l) + \boldsymbol{x}_m^T M\boldsymbol{x}_m - \boldsymbol{x}_l^T M\boldsymbol{x}_l - \tilde{o}_{ml} \right)^2$$

$$\text{s.t. } \|\boldsymbol{w}\|_2 \leq (1 - 2\rho_f)w, \ \|M\|_* \leq (1 - 2\rho_f)\mathcal{M},$$

Table 5: PI denotes percentage of pairs includes, and PA denotes prediction accuracy. Performance of logistic regression with selected pairs on HotS tournament (Hero) data. We first include the most important pairs (top tier and bottom tier) into the model, then less important pairs. In the end, we include all pairs with enough dominance information.

| PI (%) | 0 | 5.03 | 12.23 | 20.48 | 32.22 | 37.88 |
|---|---|---|---|---|---|---|
| PA (%) | 59.73 | 62.28 | 64.00 | 64.93 | 65.24 | 67.27 |

which is the problem (9) as claimed. In addition, from (14), the optimal solutions between two problems satisfy:

$$\theta^* = [\boldsymbol{w}^*, M^*] = (1 - 2\rho_f)[\tilde{\boldsymbol{w}}^*, \tilde{M}^*] = (1 - 2\rho_f)\tilde{\theta}^*$$

and the proof is thus completed.

## 9 Appendix: Another way to discover important higher order terms

In addition to Factorization HOI, another practical way to apply our basic logistic regression model (2) is to pre-select important pairwise parameters and eliminate the rest (LR-select). Intuitively, we want to identify the pairs of players that work significantly better or worse with each other. To achieve this, we may construct a tiered ranking of all possible pairs as follows.

Each game between $I_t^+$ and $I_t^-$ is "expanded" into $\binom{|I_t^+|}{2} \cdot \binom{|I_t^-|}{2}$ subgames, where every pair of players from $I_t^+$ is assigned a win over every pair of players from $I_t^-$. The results are placed into an $N_p \times N_p$ win-loss matrix, where $N_p$ is the number of observed player pairs. Winning probabilities between player pairs are now estimated using Percolation and Conductance [6], which takes advantage of transitivity of dominance information in order to infer the relationship between player pairs who may not have competed directly.

A ranking of player pairs may be obtained by applying a permutation $\rho^*$ to the rows and columns of the estimated winning probability matrix $P$ such that the cost function

$$Cost(P[\rho]) = \sum_{i=2}^{N_p} \sum_{j=1}^{i-1} \max(0, -\log[2(1 - p_{\rho,ij})]) e^{\frac{(N_p + 1 - j)(i - j)}{N_p^2}} \tag{15}$$

is minimized, as in [8]. Such a ranking can be seen in the left panel of Figure 2, in which the upper triangle of $P$ mostly consists of values above 0.5. As a result of this reordering, any deviation from $\rho^*$ incurs additional cost from cost equation (15). In particular, switching the position of a pair of entries in $\rho^*$ increases the cost by an amount dependent on their relative positions. A cost matrix $C = [c_{ij}]$ may be constructed where $c_{ij}$ is the cost incurred by swapping the $i$th and $j$th entries in $\rho^*$, an example of which is shown in the middle panel of Figure 2. By treating $C$ as a distance matrix, tiers of player pairs can then be identified via any common clustering method, such as hierarchical clustering or Data Cloud Geometry [7].

From the tiers of player pairs, we can then determine those pairs that work together much better than others (top tier) and those that work together much worse (bottom tier). We then choose to include in our model only those player pairs with very clear dominance information over other pairs, as seen in the right panel of Figure 2. Players involved in these pairs are likely to have significant relationships that make them much better or worse teammates for each other.

From Table 5 we can see that after including the most important pairs, which is only a small fraction of all pairs, the model performance improves a lot. Comparing the results with those of individual models in Table 2, we can see that by including only $12.23\%$ of pairs, LR-select is able to outperform all the other individual models on Heroes of Storm tournament data (HotS Tournament (H)).

Figure 2: Each row/column of these matrices corresponds to a pair of heroes of HotS tournament data. The estimated winning probability matrix computed via Percolation and Conductance is shown in left, and the cost incurred by swapping entries in $\rho^*$ is shown in middle. After constructing tiers of player pairs and removing pairs with little dominance information, a clearer dominance hierarchy remains regarding winning probabilities (right).