[Reviews · NeurIPS 2018]

Reviewer 1



Summary: This paper develops a model that can capture player-interactions from group comparisons (team-play win/loss info). In an effort to address higher-order interactions with a reasonable size of data set, it then proposes a latent factor model and the sample complexity analysis for the model is done under certain scenarios. Experiments are conducted on real-world on-line game datasets, comparing the win/loss prediction accuracy of the proposed approach to the prior methods such as BTL [12] and Trueskill [11]. Detailed comments: The paper studies an interesting problem, and investigates the role of player-interactions which has been out of reach in the literature. One noticeable observation found in the paper is that the proposed approach may be able to identify the best team members with good chemistry, as suggested in Table 3. Also it is clearly written. The main concerns of this reviewer lie in (a) the performance of the proposed methods with comparison to the prior works; (b) experiment settings; (c) the novelty of the proposed approach. A1. (Accuracy performance, Table 1): It seems the accuracy performance of the proposed methods is not much better than prior methods, except for HotS Tournament (H) dataset, in which players are advanced. Wondering if the other datasets reflect the cases in which players are not professional and the team chemistry is not apparent. What if the team chemistry is apparent while players are not professional? It is not clear about the scope of scenarios in which the proposed approach is outstanding – wide or narrow? A2. (Player-interaction ranking performance, Table 3): Wondering if the BTL and/or Trueskill yield similar top-5 and/or bottom-5 pairs (one can compute the team score as a simple aggregation of the estimated individuals' scores, and use them to rank player-interaction). If not, it would be good to emphasize this, as it reveals that the proposed methods can capture interaction effect which are not explored well in the prior approaches. A3. (Individual ranking performance): Many literature have focused on the ranking performance for individuals (entire or a few significant, say top-K). Wondering if the proposed approach yields the good performance for individuals as well. B1. (1st experiment setting – heroes (game characters) are individuals): This setting looks arguable because it does not capture the interaction between heroes and players – some players might be very good when playing with particular heroes. Perhaps this setting makes sense only when all players are professional and so are good enough in dealing with any character – but wondering if the setting is typical. B2. (2nd experiment setting – players are individuals): Due to the similar reason above, this setting is not quite convincing – it still does not capture the chemistry between players and heroes. I believe a reasonable justification behind the two settings should be provided – otherwise, the performance results which are based on the settings are shaking. C. (Novelty of the proposed approach): A series of the proposed approaches seem to rely on Eq. (1) in which input vector x_t is ternary: taking +1/-1 for participating individuals; 0 otherwise. But a similar approach (except the regularization term) was already proposed in the following literature: Menke-Martinez, “A Bradley-Terry artificial neural network model for individual ratings in group competitions”. I understand the proposed latent model is distinct from this, but it may be viewed as a reasonable addition to the basic framework in Menke-Martinez.

Reviewer 2



This papers proposes a pairwise version of the generalisation of BTL models for the setting of group comparisons (e.g. tournaments of teams games). Instead of considering the score of a group to be the sum of the scores of its components, this work propose to build it as the sum of the scores over the pairs of players to model interactions. Then, they add a factorization on the pairwise matrix to improve the scalability of the method and provides theoretical guarantees on the reconstruction error of this problem. The overall quality of the paper is ok. EDIT: The model is clearly explained but the positioning against feature-based methods would benefit from being more detailed, as player interactions can intuitively be learnt from feature similarities / complementarities. A logistic regression is present as a baseline, but there is very few insight about why features fail to capture the interactions, making the proposed method needed. The paper is well-organised but I have some remarks on clarifications to make: - The theoretical guarantees provided are not well-explained and discussed. It took me a certain time to understand the objective was to only to study the sample complexity and that the f^* was the loss-dependent and class-of-function dependent. The notation is a bit confusing. - The paper could be a bit more polished: l.172 I guess you mean to refer to appendix and not section 2 of the paper / please make sure all the equations are fitting the format. While the model is well-described and supported, it is a combination of familiar techniques. The BTL on the scores of the groups is already existing, the factorization method based on minimizing the nuclear norm is not new. The main novelty is to build the score of the groups as the sum over the pairs and not over the individuals. In the end, the approach seems convincing, but does not represent a major breakthrough.

Reviewer 3



This paper studies pairwise group comparison from a fixed pools of players. Based on the training data, both individual and interaction scores for each player can be learned based on a generalized linear model with potential low-rank factorization representation. It is interesting to see that such a straightforward model outperforms both previous models [11] and [12] on the datasets that the authors have tested. But it would be nice to see the comparison with more recent methods. For sample complexity analysis, assumably, the low-rank model (4) shall be better than the basic model (2). But it seems that the analysis is converted to analyzing (6), which essentially goes back to (2) with the norm constraints. It is not clear how meaningful such an analysis is. Also, there are quite a few notation inconsistency, typos, and other language errors in these parts, especially in supplement, which make these parts difficult to read. Is the problem (5) convex? If not, how initialization of SGD is done? If the global optimum can not be guaranteed, is the presented sample complexity meaningful, again? In experiments, for HotS Public (P), the basic model gets worse performance than simple logistic regression model, which seems to suggest that it is very possible that the interaction model can overfit. The authors probably should include the rank k for the factorization models for each data set. Finally, it is not really clear how Fig. 1 can reveal "interesting patterns" for player interactions. Even if the final model is rank-2, this can only tell whether the players complement each other by checking their inner product values. By the authors' response, the problem is a generalized low-rank formulation. Assuming that the convergence analysis in the provided ICML 2017 reference is correct, it does address my concern on the sample complexity analysis.